# COVID-19 Epidemic Process and Evolution of SARS-CoV-2 Genetic Variants in the Russian Federation

Vasiliy Akimkin [1,*], Tatiana A. Semenenko [2], Svetlana V. Ugleva [1], Dmitry V. Dubodelov [1] and Kamil Khafizov [1]

1    Central Research Institute of Epidemiology of Rospotrebnadzor, 111123 Moscow, Russia; uglevas@bk.ru (S.V.U.); dubodelov@cmd.su (D.V.D.)

2    Department of Epidemiology, National Research Centre of Epidemiology and Microbiology Named after the Honorary Academician N.F. Gamaleya, 123098 Moscow, Russia; meddy@inbox.ru

\*    Correspondence: vgakimkin@yandex.ru

**Abstract:** The COVID-19 pandemic, etiologically related to a new coronavirus, has had a catastrophic impact on the demographic situation on a global scale. The aim of this study was to analyze the manifestations of the COVID-19 epidemic process, the dynamics of circulation, and the rate of the spread of new variants of the SARS-CoV-2 virus in the Russian Federation. Retrospective epidemiological analysis of COVID-19 incidence from March 2020 to fall 2023 and molecular genetic monitoring of virus variability using next-generation sequencing technologies and bioinformatics methods were performed. Two phases of the pandemic, differing in the effectiveness of anti-epidemic measures and the evolution of the biological properties of the pathogen, were identified. Regularities of SARS-CoV-2 spread were determined, and risk territories (megacities), risk groups, and factors influencing the development of the epidemic process were identified. It was found that with each subsequent cycle of disease incidence rise, the pathogenicity of SARS-CoV-2 decreased against the background of the increasing infectiousness of SARS-CoV-2. Data on the mutational variability of the new coronavirus were obtained using the Russian platform of viral genomic information aggregation (VGARus) deployed at the Central Research Institute of Epidemiology. Monitoring the circulation of SARS-CoV-2 variants in Russia revealed the dominance of Delta and Omicron variants at different stages of the pandemic. Data from molecular genetic studies are an essential component of epidemiologic surveillance for making management decisions to prevent the further spread of SARS-CoV-2 and allow for prompt adaptation to pandemic control tactics.

**Keywords:** COVID-19; NGS; disease incidence; epidemic process; SARS-CoV-2; sequencing; genetic variants

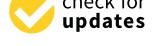



## 1. Introduction

The epidemic of a new coronavirus infection that emerged at the turn of 2019–2020, first in China and then in all countries of the world, has gone down in history as an emergency situation posing a threat to national and international security. The pandemic was caused by penetration into the human population of a new coronavirus SARS-CoV-2 (Severe Acute Respiratory Syndrome-related CoronaVirus 2), which has high contagiousness and mutational activity, as well as the absence of such a deterrent factor as baseline specific herd immunity [1–3]. The intensive development of the COVID-19 epidemic process on a worldwide scale has created favorable conditions for the emergence of genetic variants of the pathogen in accordance with the evolutionary survival strategy of viruses.

At the end of 2020, the World Health Organization (WHO) established a classification system to divide new virus variants into several subgroups: variants of concern (VOCs), variants of interest (VOIs), and variants under monitoring (VUMs), which has enabled the identification of priority areas for global monitoring, as well as the adjustment of anti-epidemic measures for COVID-19.

In March 2023, the epidemic trend tracking system and the working definitions of VOCs, VOIs, and VUMs were improved. Experts of the WHO Technical Advisory Group on the Evolution of SARS-CoV-2 (TAG-VE) reached a consensus that Omicron is currently the most divergent variant in the global landscape, undergoing mutational changes with an expanding range of circulating sublines that avoid immunologic pressure. This assertion was confirmed by the WHO announcement of the emergence and rapid global spread of new sublineages, including lineage EG.5 ("Eris"), a descendant of lineage XBB.1.9.2, as well as XBB.1.16 ("Arcturus"), XBB.1.5 ("Kraken"), and other variants of Omicron coronavirus.

Given the scale of the threat, public health systems in all countries of the world, including Russia, have set specialists the task of controlling the spread of the virus. At present, research is continuing in various areas of counteraction to the infection, such as the development of methods of diagnosis, prevention, treatment, etc. One of the important aspects of combating COVID-19 is the study of the regularities inherent in the epidemic process of this infection and the development of an epidemiologic surveillance system for the spread of new SARS-CoV-2 variants based on the information obtained.

The epidemic process of COVID-19 in each country has its own peculiarities. This is due to many factors related to the level of economic development, organization of the health care system, ethnic characteristics of the population, efficiency and volume of restrictive measures taken by the government, health and mentality of society as a whole, the state of the environment, and a number of other factors. Since the beginning of the registration of COVID-19 cases, many countries have developed and started to use statistical reporting forms for the collection of epidemiological and clinical information and the formation of databases on this infection, which have allowed for not only analyzing the features of the epidemic process and course of the disease but also assessing the effectiveness of response strategies to plan future measures aimed at containing epidemics of aerosol infections with pandemic potential [4–8].

It is important to note that all anti-epidemic measures in the Russian Federation were carried out on the basis of scientific substantiation and taking into account the experience of domestic epidemiology. In this regard, among the initial measures taken by Rospotrebnadzor (The Federal Service for Surveillance on Consumer Rights Protection and Human Wellbeing) were constant monitoring of the epidemiological situation (since 31 December 2019) and strengthening of sanitary quarantine control at checkpoints across the state border of the Russian Federation to prevent the importation and spread of cases of the disease. However, the presence of asymptomatic carriers of the virus, which are the source of infection [9], did not allow for the exclusion of the possibility of penetration of the pathogen into the territory of the country. As a result of the pandemic spread of SARS-CoV-2, the focus of the complex of anti-epidemic and preventive measures shifted from sanitary protection of the territory of the Russian Federation to laboratory testing, tracking of contacts, and their isolation within the country [10–12].

The basis for the effectiveness of the model of response to the spread of SARS-CoV-2 in Russia was the systematic and rapid introduction of timely strict restrictive measures based on the results of large-scale laboratory screening and scientific forecast of the development of epidemiological situation. The uniqueness of the approach to responding to the spread of COVID-19 in Russia is that scientific institutions are an integral part of sanitary and epidemiological services. Scientific research has become a reliable basis for the development of the means of diagnosis, prevention, and treatment of COVID-19, studying the dynamics of the epidemic process both in particular regions and in the country as a whole, which was the basis for making management decisions. It was on the basis of epidemiological analysis and clear criteria for assessing the epidemic situation that decisions were made to restrict air travel, the work of production enterprises, business organizations, educational and cultural institutions, etc. [13–15].

The aim of this study was to analyze the manifestations of the epidemic process of COVID-19 and the dynamics of circulation and rate of spread of new variants of SARS-CoV-2 on the territory of the Russian Federation. This is the first study of its kind

in the country to actively utilize coronavirus genome sequencing data throughout the COVID-19 pandemic.

## 2. Materials and Methods

Retrospective epidemiological analysis of COVID-19 incidence from 31 March 2020 to 22 October 2023 on the territory of the Russian Federation was carried out. Information about patients (age, sex, form, and date of disease) was extracted from the database formed on the basis of materials of the state official morbidity record form "Information on cases of infectious diseases in persons with suspected new coronavirus infection". These patients were assigned ICD-10 code U07.1 "COVID-19, virus identified", i.e., COVID-19 was confirmed by laboratory tests, regardless of the severity of clinical signs or symptoms. Data from the WHO, the domestic information portal Stopkoronavirus.rf2, and the Yandex DataLens3 data visualization and analysis service were also used. Based on these materials, we studied the main manifestations of the COVID-19 epidemic process for the period from the beginning of the pandemic to present day, including such characteristics as the dynamics of morbidity, gender proportion and age structure of the diseased, seasonality of morbidity, and the impact of restrictive anti-epidemic measures.

Laboratory studies were conducted after obtaining voluntary informed consent from patients with symptoms of new coronavirus infection, and the study protocol was approved by the Ethical Committee of the Central Research Institute of Epidemiology (protocol № 111 of 22 December 2020). The study used biological material obtained by swabbing from the nose, nasopharynx, and/or throat, bronchial lavage obtained by fibrobronchoscopy (bronchoalveolar lavage), (endo)tracheal, nasopharyngeal aspirate, sputum, biopsy, or autopsy material of the respiratory tract. Biological material was collected from different regions of the country, although most samples were obtained from Moscow and the Moscow region. The presence of SARS-CoV-2 RNA was confirmed by real-time reverse transcription polymerase chain reaction. RNA isolation was performed by nucleic acid precipitation using the RIBO-prep kit (AmpliSens, Moscow, Russia) according to the kit instructions. Reverse transcription was performed using the REVERTA-L reagent kit (AmpliSens, Russia) according to the manufacturer's instructions.

To analyze SARS-CoV-2 variants at different stages of the pandemic in the Russian Federation, we used sequencing data provided by the national platform for aggregating information on novel coronavirus genomes, VGARus (Virus Genome Aggregator of Russia). High-throughput sequencing was performed on the Illumina MiSeq platform (Illumina, San Diego, CA, USA) using MiSeq Reagent Kit v2 (PE 150 + 150 or PE 250 + 250 cycles) or MiSeq Reagent Kit v3 (PE 300 + 300 cycles), Illumina NextSeq 2000 using NextSeq 1000/2000 P2 reagents v3 (300 cycles), MinION using Midnight Kit (Oxford Nanopore Technologies Oxford, UK), DNBSEQ-G50 using ATOPlex RNA Library Prep Set (MGI Tech, Shenzhen,, China), and Genexus using Ion AmpliSeq SARS-CoV-2 Insight Research Assay, (Thermo Fisher Scientific, Waltham, MA, USA). Sanger sequencing of spike protein gene fragments was also performed, but this information was hardly used for detailed analysis. Nucleotide sequence data from the GISAID database were also used [16]. The Pangolin program [17], in-house tools, and scripts were used to classify SARS-CoV-2 variants.

Standard methods of descriptive statistics "Microsoft Excel 2013" and "Statistica 12.0" ("StatSoft") were used for statistical processing. The confidence interval (95% CI) was calculated using the Klopper–Pearson method (exact method).

## 3. Results

The analysis of COVID-19 epidemic process manifestations on the territory of the Russian Federation for 2020–2023 is based on the dynamic assessment of the status and trends in the development of the epidemic situation. In the Russian Federation, the first COVID-19 cases were registered on 31 January 2020 in the border areas of China. The start of the epidemic process on the territory of the Russian Federation was the importation of the first case of COVID-19 to the European part of the country (Moscow) on 2 March 2020

from Italy. The epidemic rise of disease incidence began with major metropolitan areas from 30 March 2020, and already from 16 April 2020, cases of new coronavirus infection were established in all regions of Russia. During the entire observation period (30 March 2020–22 October 2023), 23,061,960 cases of the disease were registered on the territory of the country. The average COVID-19 incidence rate in the Russian Federation in 2021–2023 was 92.9 per 100,000 population.

Dynamic assessment of the state and trends in the development of the epidemic situation of new coronavirus infection on the territory of the Russian Federation for 2020–2023 made it possible to identify two stages of the pandemic, including seven rises in morbidity. The first stage (March 2020–January 2021) is associated with the introduction of anti-epidemic and strict regime-restrictive measures of sanitary and epidemiological character in all regions of the Russian Federation, which led to a decrease in the activity of pathways of pathogen transmission. Along with nonspecific prophylaxis, the Government of the Russian Federation has consistently taken strict barrier measures: from a complete ban on entry into Russia of foreign citizens from the most affected countries to the complete closure of state borders and the termination of international air travel. Since May 2020, due to passage between susceptible persons, changes in the population harboring the new coronavirus began (increased virulence, increase in numbers), preceding the rise in incidence among the population, i.e., the process entered the phase of epidemic transformation, which was naturally accompanied by more severe cases of disease and high mortality rates. At the first stage of the COVID-19 epidemic on the territory of the Russian Federation, two rises in the incidence rate regulated by social and natural factors were recorded.

The second stage of the COVID-19 pandemic on the territory of the Russian Federation (February 2021–present) began with the change in the biological properties of SARS-CoV-2 virus with the subsequent change in prevailing (Alpha, Delta, and Omicron) variants and the start of mass specific immunoprophylaxis. The recorded five rises in COVID-19 incidence rate at stage II are probably associated with the evolution of the virus and the formation of its epidemic variant in accordance with the classical theory of the self-regulation of parasitic systems with a regular change in the immunologic structure of the human population in the chain of circulation of the pathogen [10,18]. The maximum value of the morbidity rate was recorded in the fifth period of the rise, caused by the emergence of the Omicron variant (10 January 2021–26 June 2022), amounting to 905.37 per 100,000 population (Figure 1).

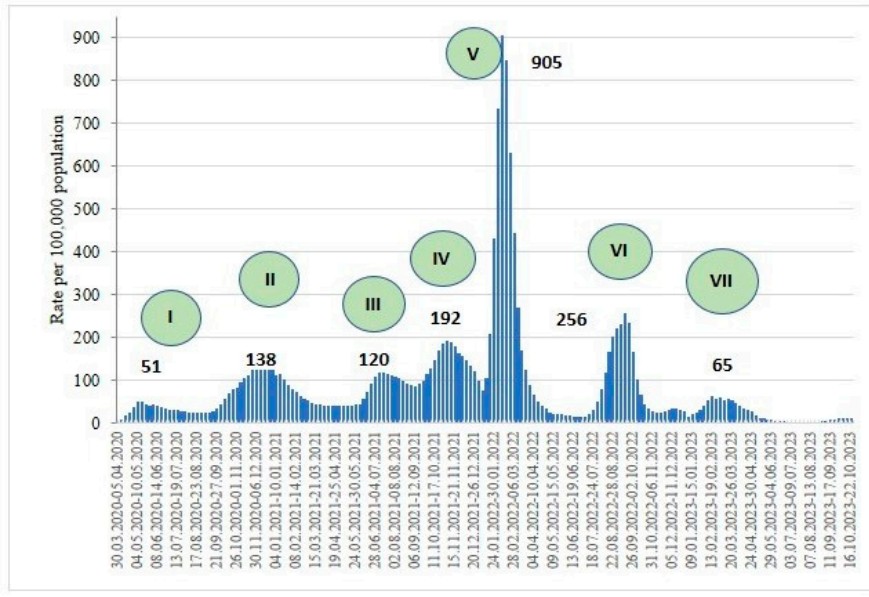

**Figure 1.** Dynamics of COVID-19 incidence in the Russian Federation in 2020–2023 (per 100 thousand population).

The most intensive spread of the SARS-CoV-2 virus was registered in three large metropolitan areas of the Russian Federation (Moscow, Moscow region, St. Petersburg), where the total percentage of cases of the new coronavirus infection during the period of its importation into the country (2 March 2020–30 March 2020) amounted to 84% (95% CI 83.08–85.2) among the total number of registered COVID-19 cases (Figure 2A).

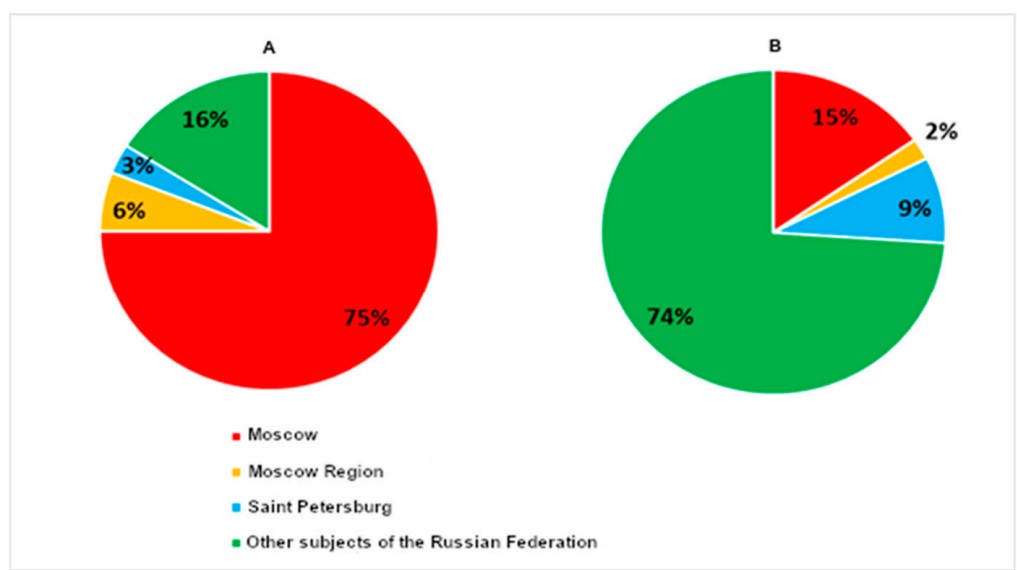

**Figure 2.** Percentage of coronavirus infection cases in large metropolitan areas (Moscow, Moscow Region, St. Petersburg) in the total structure of registered COVID-19 cases in the Russian Federation. (**A**)—during the period of SARS-CoV-2 "importation" in 2020 (2 March 2020–30 March 2020); (**B**)—between 31 March 2020 and 17 September 2023.

Subsequently, this proportion changed significantly, and the share of coronavirus infection cases in Moscow, Moscow Region, and St. Petersburg in the total structure of COVID-19 cases for the entire study period (30 March 2020–17 September 2023) amounted to 26% (Moscow—15%, Moscow Region—2%, and St. Petersburg—9%), while the share of registered cases in other regions of the Russian Federation became predominant—74% (Figure 2B).

One of the priority areas of epidemiological surveillance of COVID-19 is the identification of target population groups with the highest risk of infection. The retrospective epidemiologic analysis of data for the period 2020–2022 in different phases of epidemic development demonstrated that women and men aged 50–64 years (24.2% and 21.8%, respectively) and 65+ years (20.8% and 15.7%, respectively) constituted the majority in terms of the gender and age structure of COVID-19 patients. The lowest percentage among COVID-19 patients was observed in persons aged 18–29 years (women made up 10.9%, men 11.9%), which may be associated with the prevalence of asymptomatic forms of infection due to the active functioning of the immune system, providing effective defense of the macroorganism against infectious agents. These data coincide with the results obtained by domestic researchers in 2020, who noted that COVID-19 is a disease primarily occurring in middle-aged and older adult patients.

Despite a relatively even distribution by gender in different age groups, males predominated among COVID-19 patients under 40 years of age and females over 40 years of age (Figure 3). Thus, the demographic characteristics of COVID-19 patients over the pandemic period remain similar [19], which may indicate the relative stability of the gender and age structure. This makes it possible to define target population groups with the highest risk of infection, as well as monitoring parameters that will provide sufficient information for making targeted and effective management decisions.

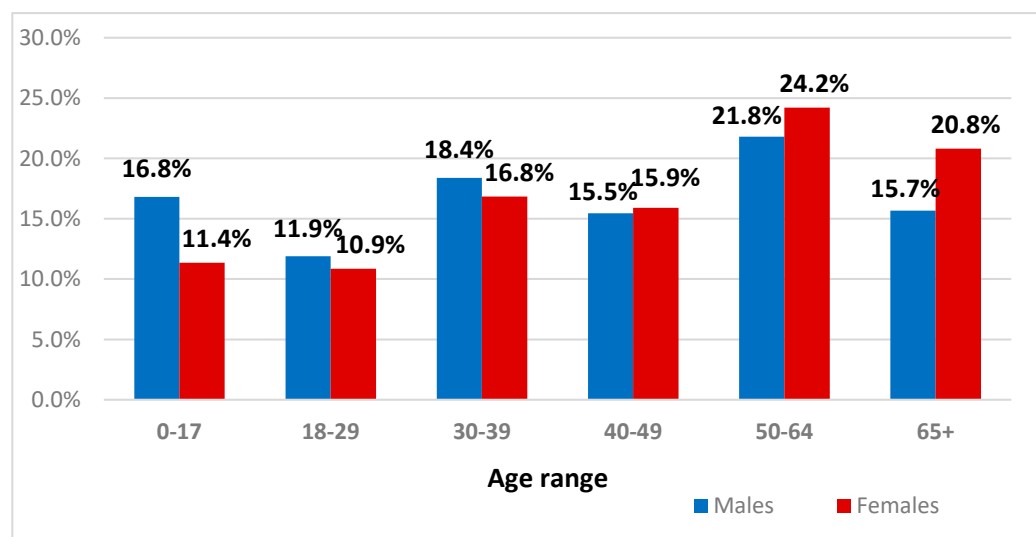

**Figure 3.** Age structure of COVID-19 cases in groups of men and women in the Russian Federation for 2020–2022.

When analyzing the incidence of COVID-19 among age groups in different periods of the pandemic, a significant increase was observed in child age groups starting from period 3. Especially worth noting are periods 5 and 6, where the age group of children under 1 year of age had the highest incidence rate—6346.8 and 3890.2 per 100,000 of the population, respectively (Figure 4).

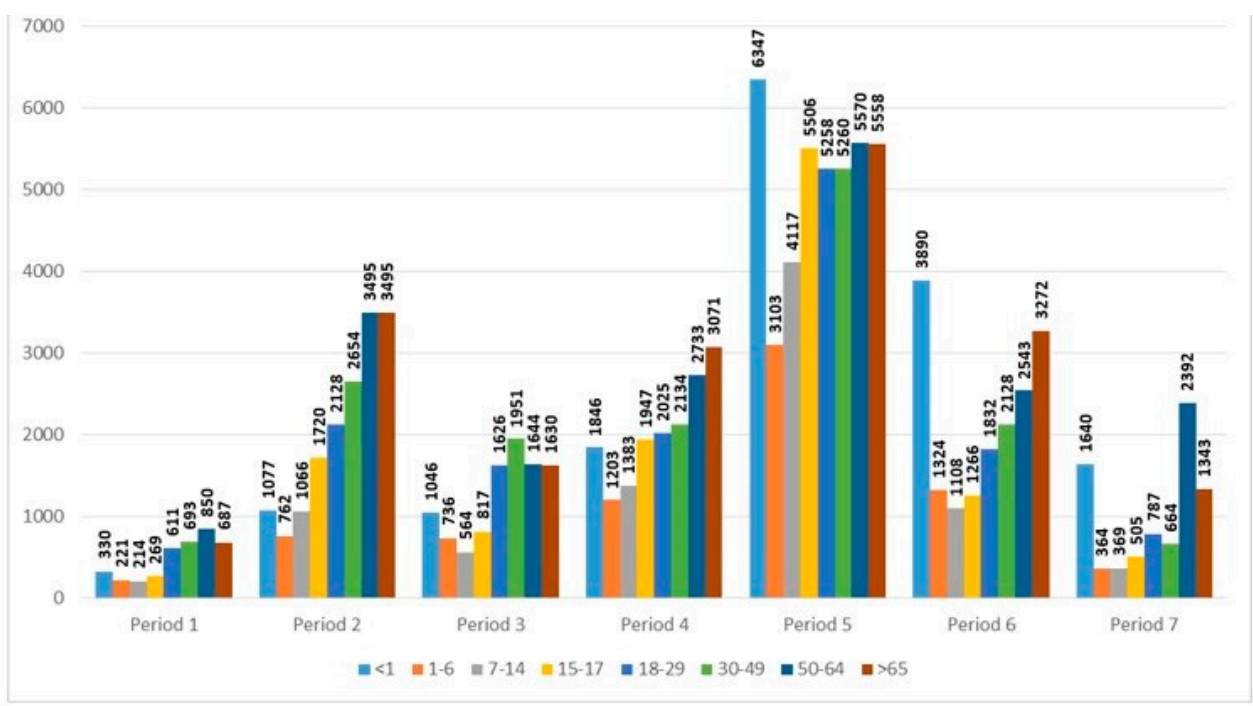

**Figure 4.** COVID-19 incidence per 100,000 of the population in the Russian Federation in age groups during different periods of the pandemic.

It should be noted that the epidemic process of COVID-19 proceeded in the pediatric population similarly to the adult population, with some lag. It was found that with each subsequent cycle of COVID-19 incidence rise, its pathogenicity decreased against the background of the increasing infectiousness of SARS-CoV-2 [20].

Intensive development of the COVID-19 epidemic process on a global scale has created favorable evolutionary conditions for the emergence of genetic variants of the pathogen, which acquire new pathogenic properties. Dynamic monitoring of mutational variability of coronaviruses detected on the territory of the Russian Federation has been carried out since December 2020, when the first case of importation (28 December 2020) of the Alpha variant (B.1.1.7) was detected. This strain, originally named "British", was subsequently renamed "Alpha" in accordance with the WHO's decision to refuse the use of countries in the names of strains. Among the mutations detected in the S-protein gene, the most significant were N501Y, P681H, and Δ69–70, which had an effect on the transmissibility of the virus, in particular, its ability to infect cells and bypass the host immune response. The detection of this variant in Russia coincided with a surge in disease incidence at the turn of 2020 and 2021.

The Beta variant (B.1.351), first detected in South Africa, and Gamma (P.1), in Brazil, were identified shortly thereafter; they were not widespread in the country, at most accounting for single percentages of the total number of new cases. The Alpha variant (B.1.1.7) was widespread in the Russian Federation in winter 2021, while the Beta and Gamma variants also occurred in early 2021.

In the summer of 2021, the appearance of the Delta variant (B.1.617.2) was accompanied by a significant increase in COVID-19 incidence and hospitalizations, which was overlaid by seasonal factors that began to decrease only towards the end of the year. The Delta gene variant spread on the territory of the Russian Federation in the second half of April 2021 and prevailed until January 2022 [21].

In November 2021, the cautious optimism of experts and hopes for a quick end to the COVID-19 pandemic ended with the emergence of a new variant of the SARS-CoV-2 coronavirus, first identified in Botswana and South Africa. On 26 November 2021, the WHO classified the mutated virus as a VOC and assigned it the code B.1.1.529, and it was named Omicron. The first subline was designated Omicron BA.1, which was quickly supplanted by another Omicron subline, BA.2, which had a significant number of differences from BA.1. The common name of the variant remained unchanged—B.1.1.529—according to PANGO classification. According to experts, SARS-CoV-2 evolved, and genomic changes led to the emergence of such characteristics as the ability to cause intensive transmission of the virus, to change the clinical symptoms of the disease, and to evade the immune response, diagnostic tools, and drugs (Figure 5).

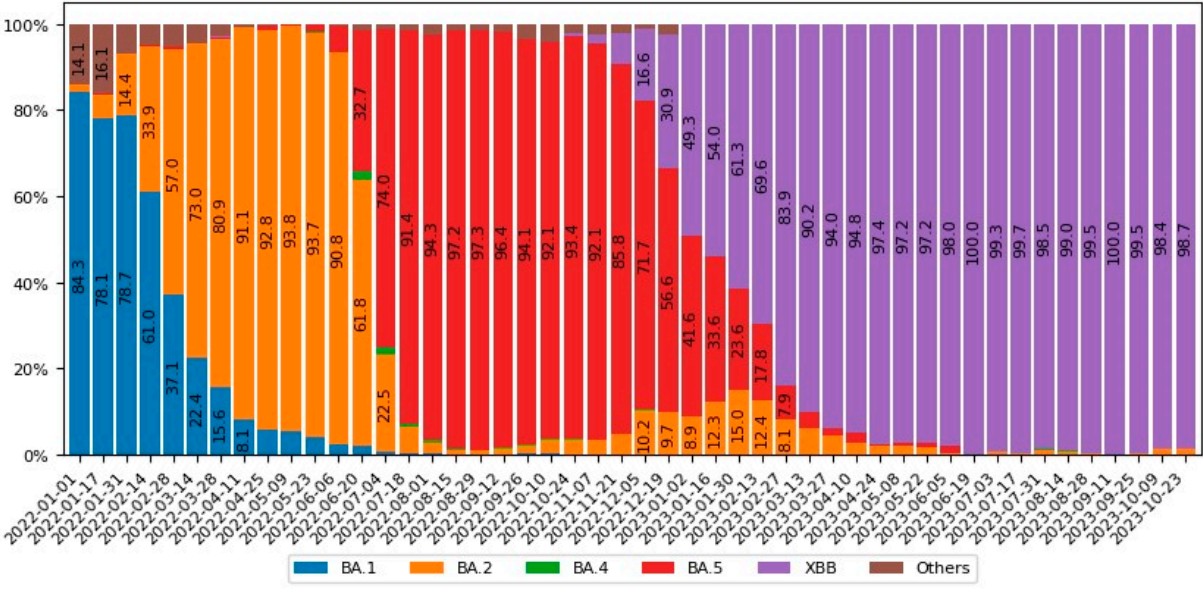

**Figure 5.** Dynamics of individual Omicron sublines from the beginning of 2022 to October 2023 in the Russian Federation.

The increase in COVID-19 incidence in the Russian Federation in early 2022 was due to the emergence of variant BA.1, which was quickly replaced by another Omicron subvariant—BA.2. Despite a period of epidemiological calm characterized by low COVID-19 incidence rates in the spring of 2022, the emergence of Omicron subvariants BA.4 and, especially, BA.5 caused a surge in incidence, which persisted until the end of October. Already near the end of 2022 and early 2023, highly transmissible variants such as BQ.1* (subvariant of BA.5) appeared, indicating the dynamic and complex nature of SARS-CoV-2 evolution. Notably, there was a resurgence of "new forms of old strains" in early 2023, particularly Omicron BA.2, which returned as recombinant forms of XBB*, dominating most of 2023. Figure 6 shows the dependence of the incidence rate of new coronavirus infections in the Russian Federation on the circulation of different SARS-CoV-2 variants.

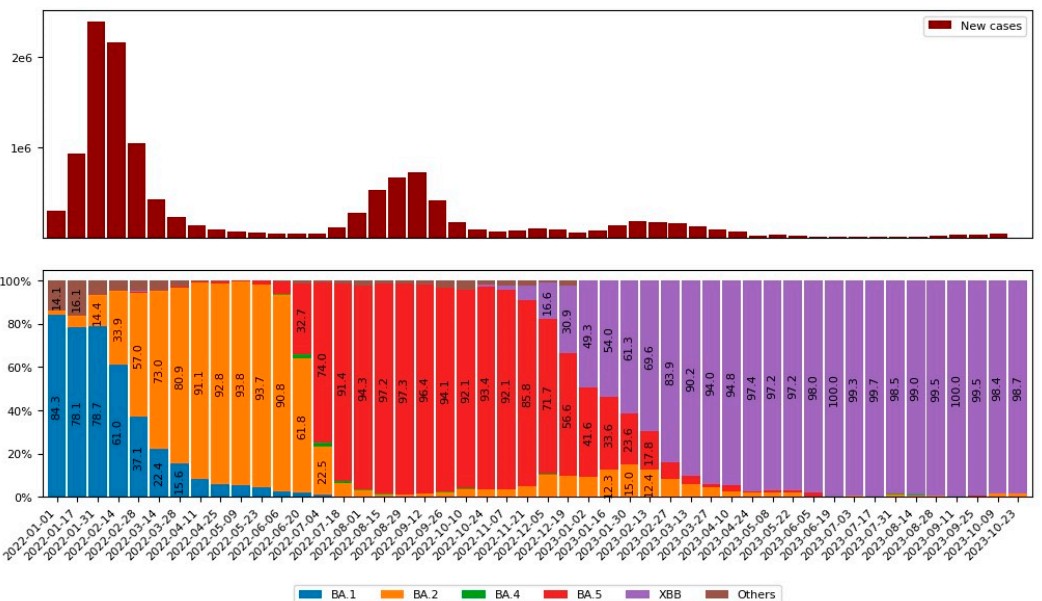

**Figure 6.** COVID-19 incidence in the Russian Federation depending on the dynamics of SARS-CoV-2 variants from January 2022 to present day.

To date, a considerable amount of data on evolutionary changes in the SARS-CoV-2 genome have been accumulated, taking into account the tendencies of acquiring new epidemiologic properties. During the period of circulation in the human population, the SARS-CoV-2 genome, adapting to a new host, acquired a certain number of nucleotide substitutions, which, among other things, is reflected in a large variety of sublines of Delta and Omicron variants uploaded to the VGARus database.

The above dynamics emphasize the need for the continuous monitoring of virus variability and genomic sequencing to detect new genetic variants in the virus population structure. The timely detection of such changes can provide a basis for the development of public health strategies and potentially help control the spread of new SARS-CoV-2 variants.

## 4. Discussion

The analysis of manifestations of the COVID-19 epidemic process on the territory of the Russian Federation distinguished two stages: at stage I (March 2020–January 2021), two rises in the incidence rate were recorded; at stage II (February 2021–present time), five rises were recorded.

The study of features of SARS-CoV-2's spread on the territory of the Russian Federation at stage I allowed the regularity of the development of the COVID-19 epidemic process to be revealed: the initial rapid rise in morbidity in the period of "importation" (2 March 2020–30 March 2020) of the pathogen in megacities due to high population density and social activity, heavy internal and international traffic flows, and small social distancing,

with a subsequent gradual involvement in the epidemic process of the population of other regions of the Russian Federation from west to east. The introduction of restrictive measures led to a decrease in the activity of pathways of pathogen transmission from the source of infection to the susceptible organism. Due to these anti-epidemic measures, there was no "explosive" growth of disease incidence on the territory of the Russian Federation, and due to the associated gain in time, it was possible to prepare medical infrastructure to provide effective professional assistance to sick people.

It should be noted that the effect of measures to separate and introduce self-isolation regimes in the conditions of a megacity comes after a time interval equal to 3.0–3.5 incubation periods with a maximum duration of 14 days [22]. Against the background of the self-isolation regime, the turning point in the development of the COVID-19 epidemic in Moscow occurred on 16 May 2020, when the first significant decrease in the number of new cases occurred from 4748 to 3505, with stabilization at the achieved level, and a subsequent decrease was recorded.

Stage II of the COVID-19 pandemic in the Russian Federation (from February 2021 to the present) began with a change in the biological properties of the SARS-CoV-2 virus with the subsequent change in the prevailing (Alpha, Delta, and Omicron) variants and the start of immunoprophylaxis. Rises in COVID-19 incidence occurred against the background of mass vaccination and are probably associated with the evolution of the virus and the formation of its epidemic variant (phase development of the epidemic process) with natural changes in herd immunity during the circulation of the pathogen [18]. In addition, the rise in the incidence of COVID-19 was presumably associated with seasonal factors characteristic of the increase in the incidence of respiratory infections with the airborne transmission of the pathogen. At present, the materials available for the retrospective long-term analysis of COVID-19 incidence are insufficient. It can be assumed that the coronavirus acquires its seasonality, with an annual rise in incidence in September–October, but these data require further study and confirmation.

The COVID-19 pandemic in the Russian Federation, as well as in other countries of the world, has become a serious test for health care systems and sanitary and epidemiologic surveillance due to the increased demand for laboratory tests and the need to increase the throughput capacity of medical and research laboratories. In this context, the Central Research Institute of Epidemiology of Rospotrebnadzor produced reagent kits for different stages of analysis using nucleic acid amplification methods for more than 85 million tests during the pandemic period. A platform for the monitoring and analysis of information on the results of COVID-19 testing was developed, within which all organizations transmitted information automatically online within 2 h using API integration, with the use of cryptographic protection. The received data were fed into the secure system of laboratory aggregation results (SOLAR), to which more than 1800 treatment and prevention organizations of Rospotrebnadzor, Ministry of Health, as well as network and local laboratories, were connected. For the entire period of the system's operation, starting from 1 November 2020, more than 180 million diagnostic test results were received in total. It is important to note that the flexibility of the SOLAR system makes it possible to customize the platform in the shortest possible time and start aggregating data for other significant diseases as well, which will allow us to promptly respond to new foci of morbidity and prevent the spread of morbidity in different regions of the Russian Federation [23].

Like other RNA viruses, SARS-CoV-2, adapting to its new human hosts, is subject to genetic evolution, which leads to mutations in the viral genome that can alter the pathogenic potential of the virus. The most effective approach for molecular genetic monitoring of pathogen variability is next-generation sequencing (NGS) technologies, which allow for comprehensive information to be obtained about the pathogen genome and the appearance of new mutations to be traced [23,24]. The key and most significant example of the application of molecular genetic monitoring was the detailed study of coronavirus during the COVID-19 pandemic. By analyzing the genomes of SARS-CoV-2, it was possible to establish links between different variants of the pathogen and the peculiarities of the

epidemic process. The analysis of significant amounts of data and the use of modern methods makes it possible to accurately monitor the epidemic situation, understand the relationship between genetic variants of viruses and their pathogenicity and infectiousness, and take timely and targeted proactive measures to prevent the spread of infection.

A major technological advance during the SARS-CoV-2 pandemic was the development of a lineage classification system that provides a relatively universal, high-resolution picture of SARS-CoV-2 genetic diversity. Together with the related Pangolin tool [17], researchers, clinicians, and policy makers gained a common language to discuss SARS-CoV-2 genetic diversity without having to generate or interpret a phylogeny. In 2020, other classification systems were introduced, including the Nextstrain, which aimed to better characterize phylogeny.

In the Russian Federation, in accordance with the Decree of the Government of the Russian Federation, the Russian platform for the aggregation of information on virus genomes "VGARus", which contains a large set of SARS-CoV-2 sequences and represents an invaluable resource for deciphering the development of the COVID-19 pandemic, has been developed and put into operation at the Central Research Institute of Epidemiology of Rospotrebnadzor. More than 150 organizations are currently integrated into the system, with a significant proportion of them performing the mass sequencing of SARS-CoV-2 genomes and uploading sequences to VGARus for further analysis. Each sample in the system not only contains nucleotide sequence and technical data but also includes information on the place and time of the collection of biological material, as well as data on the examined person: sex, age, vaccination status, estimated number of contacts, comorbidities, recent foreign travel, etc. These data, given their epidemiologic significance, can be used in future studies as well.

It should be emphasized that at the present stage, the epidemic process of COVID-19 in the world is in a state of unstable dynamic equilibrium, and even a slight increase in the transmissibility of the pathogen, under the same other conditions, can lead to an increase in incidence [3]. The evolution of the virus does not stop, and within the XBB lineage, its own "leaders" appear, for example, XBB.1.5 ("Kraken"), XBB.1.16 ("Arcturus"), and XBB.1.9.2.1 (EG.5, "Eris"). The emergence of the latter coincided with the beginning of the increase in the incidence of the disease in the country in September 2023. Finally, at the end of August, BA.2.86 (informally named "Pirola"), a new variant of SARS-CoV-2, was discovered in Israel and Denmark, which has a number of additional mutations compared to previously identified Omicron variants. Specifically, the genetic sequence of BA.2.86 differs by more than 30 amino acid substitutions from BA.2. BA.2.86 also has >35 amino acid substitutions compared to circulating variant XBB.1.5 ("Kraken"), which is dominant through most of 2023. This number of genetic differences roughly corresponds to the number of mutations between the original Omicron variant (BA.1) and previous variants such as Delta (B.1.617.2). BA.2.86 was designated by the WHO as a variant under monitoring on 17 August 2023, and as a variant of interest in November 2023. As of the end of November 2023, more than 10,000 BA.2.86 sequences have been officially registered worldwide (GISAID data). It is important to note that the amount of genomic sequencing of SARS-CoV-2 in the world has significantly decreased compared to previous years (about 10-fold compared to August 2022), which means that new variants may emerge and spread without being detected for a long time.

## 5. Conclusions

In accordance with the decision of the Committee on Coronavirus Infection on 5 May 2023, the WHO declared that COVID-19 no longer constituted an emergency of international concern. However, the Committee's position has changed over the past few months. Experts continue to recognize the continuing uncertainty associated with the potential evolution of the virus and the emergence of multiple SARS-CoV-2 variants [25]. In this regard, all countries, including the Russian Federation, are recommended to strengthen epidemiologic surveillance, to actively conduct genomic sequencing for the effective tracking

of circulating SARS-CoV-2 variants, and to deposit full sequences of the virus genome and associated metadata in a publicly available database, such as GISAID. Due to the global dominance of the Omicron variant, it is necessary to study its impact on the severity of the disease, the effectiveness of anti-epidemic measures, immune response, neutralizing antibody activity, and other parameters of interest.

**Author Contributions:** Conceptualization, V.A., T.A.S. and K.K.; methodology, V.A., T.A.S., S.V.U., D.V.D. and K.K.; software, D.V.D. and K.K.; validation, V.A., T.A.S., S.V.U., D.V.D. and K.K.; formal analysis, V.A., T.A.S., S.V.U., D.V.D. and K.K.; investigation, V.A., T.A.S., S.V.U., D.V.D. and K.K.; resources, V.A.; data curation, V.A., T.A.S., S.V.U., D.V.D. and K.K.; writing—original draft preparation, V.A., T.A.S., S.V.U. and K.K.; writing—review and editing, V.A., T.A.S., S.V.U., D.V.D. and K.K.; visualization, T.A.S., S.V.U., D.V.D. and K.K.; supervision, V.A.; project administration, V.A.; funding acquisition, V.A. All authors have read and agreed to the published version of the manuscript.

**Funding:** The work on sequencing and data analysis was carried out at the expense of a number of subsidies from the Government of the Russian Federation and own funds of the Central Research Institute of Epidemiology of Rospotrebnadzor.

**Institutional Review Board Statement:** The study protocol was approved by the Ethical Committee of the Central Research Institute of Epidemiology (Protocol No. 111 of 22 December 2020).

**Informed Consent Statement:** Informed consent was obtained from all subjects involved in the study.

**Data Availability Statement:** VGARus platform data are available for authorized users, after signing an agreement. Some coronavirus genome sequencing data in the Russian Federation are freely available for download from the GISAID database.

**Conflicts of Interest:** The authors declare no conflicts of interest.

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
