# Peer review of "COVID-19 Epidemic Process and Evolution of SARS-CoV-2 Genetic Variants in the Russian Federation"

_2036-7481, doi:10.3390/microbiolres15010015_

Round 1
Reviewer 1 Report
Comments and Suggestions for Authors
The manuscript describes the historical series of COVID-19 in Russia, highlighting genetic variants throughout the period. It is an interesting record, which compiles the history of the disease in that country. Some questions follow below:
PAG 6 LINE 206
a - during the period of SARS-CoV-2 "importation" in 2020 (02.03.2020 - 30.03.2020); b - between 30.03.2020 and 17.09.2023. 03/30/2020 appears in both intervals. Doesn't interval B start on 03/31/2020?
Pag 9 line 314
4. Discussion
analysis of manifestations of the COVID-19 epidemic process on the territory of the Russian Federation allowed to distinguish two stages: at the I stage (March-January 2021) two rises of the incidence rate were recorded, at the II stage (January 2021 - present time) - five rises were recorded.
- at the I stage (March-January 2021) change to - at the I stage (March 2020-January 2021)
- at the I stage (March-January 2021) two rises of the incidence rate were recorded, at the II stage (January 2021 - present time) - five rises were recorded. January 2021 appears in both intervals. Doesn't (February 2021 - present time)? the same period/date cannot appear in two intervals, otherwise they will be counted in duplicate
Author Response
We would like to thank the reviewer for the positive opinion of our article. The responses are summarized below item by item.
PAG 6 LINE 206
a - during the period of SARS-CoV-2 "importation" in 2020 (02.03.2020 - 30.03.2020); b - between 30.03.2020 and 17.09.2023. 03/30/2020 appears in both intervals. Doesn't interval B start on 03/31/2020?
Yes, we agree with the comment. The importation of the pathogen into the Russian Federation occurred from 02.03.2020 to 30.03.2020, and the first rise in the incidence of the disease was from 31.03.2020 to 17.09.2020. We have made corrections to the text.
Pag 9 line 314
- Discussion
analysis of manifestations of the COVID-19 epidemic process on the territory of the Russian Federation allowed to distinguish two stages: at the I stage (March-January 2021) two rises of the incidence rate were recorded, at the II stage (January 2021 - present time) - five rises were recorded.
- at the I stage (March-January 2021) change to - at the I stage (March 2020-January 2021)
- at the I stage (March-January 2021) two rises of the incidence rate were recorded, at the II stage (January 2021 - present time) - five rises were recorded.
January 2021 appears in both intervals. Doesn't (February 2021 - present time)? the same period/date cannot appear in two intervals, otherwise they will be counted in duplicate
We agree with the comment. Phase I-March 2020-January 2021; Phase II-February 2021-present. Corrected in the text now.
Reviewer 2 Report
Comments and Suggestions for Authors
Author Response
We would like to thank the reviewer for his positive opinion of our work and his questions and comments.
- The quality of Figures 3 and 4 is quite poor. The quality of these images can be improved.
Thank you for the observation. Figures 3 and 4 have been corrected to improve the quality.
- Several other review studies on this subject have previously been published. What is the importance of this research, which requires more clarification and emphasis?
We thank the reviewer for the comment. This is the first study on the epidemiology of a novel coronavirus infection in the Russian Federation that actively used genomic data from the VGARus database to explain the course of the pandemic in the country. We have noted this in the text (end of the Introduction section).
- The manuscript has grammatical and typo errors which require more refinement.
Thank you for the observation. We have carefully gone through the text and corrected all the errors we noticed.
- Some citations are not consistent. Consistency in referencing style is essential.
We carefully went through the list of citations, and corrected to conform to a uniform style.